# Measuring the Autistic Women’s Experience (AWE)

**DOI:** 10.3390/ijerph20247148

**Published:** 2023-12-06

**Authors:** Yvonne Groen, W. Miro Ebert, Francien M. Dittner, Anne Fleur Stapert, Daria Henning, Kirstin Greaves-Lord, R. C. D. (Lineke) Davids, Stynke Castelein, Simon Baron Cohen, Carrie Allison, Ingrid D. C. Van Balkom, Sigrid Piening

**Affiliations:** 1Clinical and Developmental Neuropsychology, University of Groningen, 9712 TS Groningen, The Netherlands; 2Institute for Sport Sciences, University of Regensburg, D-93053 Regensburg, Germany; 3Autism Team Northern-Netherlands, Jonx, Department of (Youth) Mental Health and Autism of Lentis Psychiatric Institute, 9728 JR Groningen, The Netherlands; af.stapert@lentis.nl (A.F.S.); idc.vanbalkom@lentis.nl (I.D.C.V.B.); s.piening@lentis.nl (S.P.); 4Lentis Psychiatric Institute, Outpatient Clinic for the Elderly, 9725 AG Groningen, The Netherlands; daria.henning@ggze.nl; 5Clinical Psychology and Experimental Psychopathology, University of Groningen, 9712 TS Groningen, The Netherlands; 6Martini Hospital, Medical Psychology, 9728 NT Groningen, The Netherlands; l.davids@mzh.nl; 7Lentis Research, Lentis Psychiatric Institute, 9725 AG Groningen, The Netherlands; 8Autism Research Centre, Department of Psychiatry, University of Cambridge, Cambridge CB2 8AH, UK

**Keywords:** autism, screening, factor analysis, female, sensory sensitivity

## Abstract

We developed a Dutch questionnaire called the Autistic Women’s Experience (AWE) and compared its psychometric properties to the Autism Spectrum Quotient (AQ). Whilst attenuated gender differences on the AQ have been widely replicated, this instrument may not fully capture the unique experience of autistic women. The AWE was co-developed with autistic women to include items that reflect autistic women’s experience. We investigated the AWE (49 items) and compared it with the AQ (50 items) in Dutch autistic individuals (*N* = 153, *n* = 85 women) and in the general population (*N* = 489, *n* = 246 women) aged 16+. Both the AQ and AWE had excellent internal consistency and were highly and equally predictive of autism in both women and men. Whilst there was a gender difference on the AQ among non-autistic people (men > women), there was no gender difference among autistic people, confirming all earlier studies. No gender differences were detected on the AWE overall scale, yet subtle gender differences were observed on the subscales. We conclude that the AQ is valid for both genders, but the AWE provides an additional useful perspective on the characteristics of autistic women. The AWE needs further validation in independent samples using techniques that allow for testing gender biases, as well as a confirmatory factor analysis in a larger sample.

## 1. Introduction

### 1.1. Relevance and Study Aim

The clinical diagnosis of autism in females is often delayed or misdiagnosed compared to autism in males [1,2,3,4,5,6]. Delayed diagnosis is associated with several negative outcomes, such as reduced societal participation [7], impaired quality of life [8,9], worse mental health [10] and a higher risk of being a victim of sexual abuse [11]. Autistic women often experience a strong drive to conform with societal expectations and rules despite their social difficulties and therefore use coping strategies to pretend to be ‘normal’ [3]. Pretending to be ‘normal’, also called masking or camouflaging, is more common in autistic women than in autistic men [12]. Camouflaging or masking often leads to exhaustion, increased mental distress and feelings of disempowerment, particularly in those with a delayed diagnosis [13]. This results in autism being less well integrated into the individual’s identity and being less accepted by both the person and others around them. Reduced acceptance by others as well as reduced personal acceptance of being autistic is linked to depression [14]. Earlier identification may improve timely access to relevant services, which may lead to a healthier coping style and self-acceptance. It is therefore important that diagnostic instruments for autism are relevant to both men and women.

Lai and colleagues [15] called for more research into sex and gender differences in autism, for a better understanding of autism in women and for less male-biased research. This research should not only pertain to biological mechanisms that may be implicated differently between the sexes (such as a potential higher liability threshold in females) [16] or differences in prenatal sex hormones (steroid and oestrogens) [17,18], but should also investigate similarities and differences between the genders in social behaviour and autistic traits [15]. Being aware of similarities and differences between genders may change the way autism is defined, recognised and diagnosed and ultimately lead to fewer autistic women experiencing mental health distress. Autistic women may be less easily recognized compared to autistic men because they show different traits than those traditionally associated with autism. It is therefore important to document a wider range of autistic behaviours that are more descriptive of autistic females, beyond the existing (male-biased) instruments [4,15].

The current study aimed to identify such female-specific autistic behaviours and to develop a new instrument for autistic women: the Autistic Women’s Experience (AWE). The AWE goes beyond the Autism Spectrum Quotient [19] by adding items describing autistic traits that were suggested by autistic women. Thus, this study addresses a gap in existing diagnostic instruments for autistic women. This new instrument may help to address the male bias in autism screening instruments and therefore optimize gender sensitive care, by recognizing that women and men may experience mental health issues differently.

### 1.2. Gender Differences in Autism

Historically, the terms sex (as a biological construct) and gender (as a social construct) are often conflated [20]. In this article, we consistently use the term ‘gender’, because we are writing about attitudes, feelings and behaviours that our culture associates with autistic women and men. Only when we write about specific biological correlates of autism do we use the term ‘sex’. Differences between sex and gender have largely been neglected in clinical research, although it is known that they can affect outcomes such as prognosis or response to treatment [21]. A frequent assumption in research is that it is sufficient to study medical or psychological conditions in one gender and that the results will be applicable to the other [22]. In autism, this idea was disputed as early as 1981 [23]. In the early 2000s, the gender ratio was 4:1 (male:female) [24] and as high as 10:1 (m:f) in those with an average or above average IQ [25,26]. This may in part have contributed to the under-representation of females in autism research. Due to the increase in diagnoses among females and a greater awareness of camouflaging and under- or misdiagnosis, the gender ratio is now closer to 3:1 (m:f) [27]. Increases in diagnoses are reported to be greater for females than males [28]. Currently, sex and gender differences in autism are an important topic of debate, because our current understanding of autism may rely too heavily on the clinical presentation of autism in males [4,15,29].

Considerable gender differences have been observed in autism characteristics [4,15]. The core symptom, ‘deficits in social communication and social interaction’, often manifests more subtly in autistic women compared to autistic men [30], particularly when focusing on DSM core characteristics [31]. Autistic girls and women present with more advanced social interaction and communication than autistic boys and men, which mirrors gender differences in the general population. However, according to autistic women, the impact of their autistic traits on their wellbeing is often very damaging. They indicate that their efforts of trying to hide socially unconventional behaviours (camouflaging) has a severe impact on their mental wellbeing [13,32,33]. Using the Camouflaging Autistic Traits Questionnaire (CAT-Q), we found that autistic women more often camouflage their autism than do autistic men [12]. Consciously or unconsciously, autistic women are more likely to camouflage their difficulties through masking (e.g., monitoring and adjusting the face and body to appear relaxed) and assimilation (e.g., forcing oneself to interact with others and pretending to be non-autistic), although they do not use more compensation strategies (e.g., using scripts in social situations or researching the rules of social interactions) in comparison to autistic men.

Furthermore, greater social motivation is observed in autistic girls compared to autistic boys, as measured by the social responsiveness scale (SRS) [34]) completed by parents, as well as teacher ratings [35]. A combination of being highly socially motivated yet having lower social skills may hamper them in developing friendships and relationships [36]. For example, autistic girls are less capable of identifying relational conflict than neurotypical girls [35]. Compared to autistic boys, autistic girls may be better at initiating friendships, but may have more difficulties maintaining these friendships [37]. The core symptom of ‘repetitive and restricted patterns of behaviour’ is significantly less commonly present, less pronounced and different in presentation in autistic women compared to autistic men [38,39]. Autistic women and girls may have more socially accepted special interests. Whereas autistic boys might show great interest in, for example, license plates, house numbers or trains, autistic girls tend to have preoccupations that are more socially oriented (such as reading, other girls, and romantic relationships) and more accepted as ‘normal’ (such as dolls or horses) [40,41]. Unusual sensory responses are often reported to be more acute in autistic girls compared to boys [42,43].

### 1.3. Identification of Autism in Females

The current male bias in understanding autism may hinder the effective identification of autistic females. Girls with signs of autism tend to be referred and diagnosed later than males [5,6]. Begeer and colleagues [1] reported a diagnostic delay ranging from 1.8 years to 4.3 years in Dutch women versus men. One reason for this delay could be due to the greater use of strategies to appear less autistic (camouflaging) [44,45]. It is perhaps more difficult for parents and teachers to identify autism in girls, partly due to outdated myths about autism, for example, that autism is a male condition. In addition, social withdrawal may be seen as shyness in girls, but as an autistic trait in boys [41], which may make it less likely for girls to be identified. Furthermore, clinicians may misdiagnose other conditions in females, e.g., depression, obsessive-compulsive disorder, social anxiety, eating disorders or borderline personality disorder [40], and perhaps have internalised different diagnostic thresholds for girls and women compared to boys and men. Additionally, current autism screening and diagnostic instruments may be less able to detect autistic females because the instruments are biased towards the clinical presentation of autism in males [46], and norms based on predominantly male samples contribute to a reduced sensitivity to autism in females [29].

The Autism Spectrum Quotient (AQ) [19] is a 50-item screening questionnaire measuring autistic traits in adults (with average to above average intellectual abilities). Internationally, it is the most widely used (open access) measure of autistic traits in research and clinical practice. Several studies evaluating the full-scale scores of different versions of the AQ (AQ-short, adolescent AQ, adult AQ) suggest that they are sensitive to the presence of autistic traits, irrespective of gender [19,47,48]. The Dutch full-scale AQ performs well in predicting autism and discriminating between autism and other conditions; in clinical practice, the AQ has higher indices of sensitivity and specificity than those of the SRS Adult Version [49]. However, inconsistencies have been noted in the factor structure (structural validity) of the AQ in the literature, including relatively poor fit indices for the original five-factor model [47,50,51,52,53] and less than satisfactory internal consistency reliability (Cronbach’s alpha, α < 0.7) for each factor in the original AQ [50,53,54,55]. Regarding gender differences on the full-scale AQ, results from a big data study (including over 800 autistic men and women) reported small gender effects within the autism group, indicating that typical gender differences (typically of a medium effect size) are attenuated in autistic men and women [56]. Very few autistic women were included in the first validation study of the AQ (13 women vs. 45 men) [19]. Later, subsequent big data studies have tested the AQ in populations as large as half a million people in the general population [57] and a short version of the AQ in 36,000 autistic people [58]. Contemporary female-specific autistic characteristics that have been more often observed in women in the past decade [4,15,59] were not included in the original AQ. For example, the AQ does not cover camouflaging behaviours, difficulties with maintaining friendships or more socially accepted interests, which are all experienced by autistic women. Equally, the AQ does not include altered responses to sensory stimuli, which have been added to the diagnostic criteria in DSM-5 [30] and may also be more common in autistic women [60]. Even though the Camouflaging Autistic Traits Quotient (CAT-Q) [12] and the Sensory and Perception Quotient (SPQ) [61] were developed to cover these issues, to date, no screening questionnaires are available that capture the full range of autistic characteristics that appear to be more common among autistic women.

### 1.4. Current Study

We propose that an additional instrument is needed that (1) better reflects the full range of characteristics that might be more common among autistic women; (2) includes autistic characteristics that matter to women with autism; (3) has high sensitivity and specificity in the assessment of autistic women; and (4) provides gender-specific norms. This may help overcome the diagnostic bias towards men and the under-identification or misdiagnosis of autistic women since women may not endorse items that are typically male. We developed the Autistic Women’s Experience (AWE) and co-created its items with autistic women, since the involvement of experts by experience is fruitful in the research process and in the development of questionnaires [62]. The AWE is timelier than the AQ since the AQ was designed two decades ago, when the understanding of autism was predominantly male-focused. Whilst attenuated gender differences on the AQ have been widely replicated, including in big data, the recent literature on autistic women suggests that the traditional expression of autism may not fully capture their unique experience. The current study describes the development and psychometric properties of the AWE with the aim to successfully discriminate between autistic and non-autistic women. The new items were developed in the Dutch language by our multidisciplinary team (EmFASiS) in collaboration with ‘experts by experience’ (autistic women), based on current knowledge in the literature, a recent Delphi consensus study [4] and qualitative studies [63] of the EmFASiS members.

The current study investigates the psychometric properties of the AWE, tests its sensitivity and specificity for autism in women and men, and makes a direct comparison of the test properties with the original Dutch AQ. We hypothesized that autistic women would score higher than autistic men on the AWE and that both autistic men and women would score higher than gender-matched non-autistic people. Additionally, we hypothesized that the sensitivity and specificity of the AWE would be better in women than in men, whereas the sensitivity and specificity of the AQ would be better in men than in women.

## 2. Materials and Methods

### 2.1. Participants

Two groups of Dutch women and men over the age of 16 years were recruited. One group consisted of *N* = 153 people with a clinical autism diagnosis, according to the DSM-5 criteria (American Psychiatric Association, 2013), and the other group was a general population sample (*N* = 489) of people without a diagnosis of being on the autism spectrum. Table 1 contains an overview of demographic characteristics of the samples, separated by gender and diagnosis, including age, age of diagnosis, living situation, highest level of education obtained, enrolment in an education programme and job status. Participants who indicated they were of “other gender” were excluded from the analysis, since our research question focused on differences between men and women. The “other gender” group was too small to allow for subgroup analyses.

The autism group was recruited from a specialised outpatient clinic for autistic individuals in the Netherlands. All eligible individuals registered at this centre were contacted via email to participate in this research online. In total, 1017 invitations were sent, and the response rate was 17% after sending out two reminders. Patients were excluded if they did not consent to participate in the study, were aged under 16 years, indicated they did not have a current diagnosis of autism, and did not pass all control questions (see Section 2.2). Only 3% of the patients were excluded based on careless responding, as checked by the control questions (e.g., “To check whether you are still paying attention, please choose slightly agree”). In total, 145 autistic people were included, of which 79 were women and 59 were men. The mean age was 39 years (SD = 14). Thirty percent of the autistic participants had received their diagnosis before the age of 18 years. The variation in the age of diagnosis according to each gender is described in the results section (Section 3.5). More than half of the people diagnosed as autistic (57%) indicated they had one or more secondary mental health conditions, including post-traumatic stress disorder (*n* = 19), anxiety disorders (*n* = 32), depression or affective disorders (*n* = 44), an eating disorder (*n* = 8), a personality disorder (*n* = 8), a psychotic disorder (*n* = 2) or attention deficit hyperactivity disorder (ADHD) (*n* = 4).

The general population sample was recruited via two routes. One route was through social media and personal networks, in which 324 Dutch adults aged 16 years or older participated in the study, without compensation. The other route was through a company for online research (Panel Inzicht), in which 344 Dutch adults aged 16 years or older were recruited, receiving financial compensation for their participation. Participants were excluded from the dataset if the survey was incomplete (11%), if they indicated to have an autism diagnosis (8%), if they had extremely short response times of less than 5 min (1%) and if they did not pass the control questions (7%). The final sample consisted of 477 individuals, of which 236 were women, 238 were men, 2 (0.4%) indicated they were ‘other gender’ and 1 (0.2%) did not disclose their gender. The mean age of the general population sample was 43 years (SD = 18). Seventy-seven participants (16.1%) indicated to have one or more mental health or neurodevelopmental diagnosis other than autism, including post-traumatic stress disorder (*n* = 15), anxiety disorders (*n* = 20), depression or affective disorders (*n* = 27), an eating disorder (*n* = 4), a personality disorder (*n* = 11), a psychotic disorder (*n* = 2) or attention deficit hyperactivity disorder (ADHD) (*n* = 10). 

To measure test/retest reliability, a convenience sample was recruited from the social network of the EmFASiS team, who completed the questionnaires twice with an average interval of 8 weeks (range of 5–12 weeks, SD = 2.1). This sample consisted of *N* = 70 participants (*n* = 37 women) with a mean age of 41 years (range of 18–82 years, SD = 18.9 years).

### 2.2. Materials

#### 2.2.1. Autistic Women’s Experience (AWE)

To develop the AWE, we created 52 new Dutch items to measure autism in women (see Appendix A for the full-length list of all items used in this study). Drafting the items was a team effort by the multidisciplinary EmFASiS research team (emfasisonderzoek.org accessed on 4 December 2023) consisting of psychologists, sociologists, a psychiatrist, a linguist and a medical doctor, in collaboration with experts by experience (including members of the EmFASiS sounding board group). The new items were written in the same style as the AQ items and used the same Likert response scale and scoring system as the Dutch AQ (see Section 2.2.2). We followed the item writing procedure as recommended by the guidelines for scale development that have been formulated by Clark and Watson [64]. First, we performed literature searches on the presentation of autism in females [4], and then we conducted qualitative interviews with autistic men and women. Further, we conducted a Delphi consensus study combining the perspectives of clinicians and scientists about a female autistic phenotype [4]. We added new female-specific items to extend the original factors of the AQ particularly for the scales “social skill”, “communication” and “attention to detail”. Finally, we added items enquiring about sensory processing which we expected to form a separate subscale. Our hypothesis was that sensory processing would emerge as an isolated construct (i.e., “orphan construct”) in the questionnaire with lower item total correlations, similar to the existing AQ “attention to detail” subscale, with low intercorrelations with other factors, but still adding predictive power to the construct of “autism” [64]. In total, we intended to create six subscales. Out of the 52 new items, 32 were reverse-scored. For the purpose of this publication, two professional translators translated and back-translated the items into English. Only minor discrepancies were found between the Dutch items and back-translated items, and these were solved between the translators and SP/YG.

#### 2.2.2. Autism Spectrum Quotient (AQ)

The AQ includes 50 items which measure traits associated with the autism spectrum [19]. See Appendix A for the full-length list of the original items that were used in this study. The Dutch AQ was used in this study, which was previously translated with a back-translation procedure into Dutch [47]. Responses are given on a 4-point Likert scale (1 = “definitely agree”, 2 = “slightly agree”, 3 = “slightly disagree” and 4 = “definitely disagree”). Roughly half of the items (24 of 50) are reverse-scored (i.e., “definitely agree” scores 4 points and “definitely disagree” scores 1 point). Unlike the original binary item scoring of the AQ, all Likert item scores are summed in the Dutch AQ [47], with a minimum score of 50 and a maximum score of 200. Higher scores on the AQ indicate greater presence of autistic characteristics. Hoekstra and colleagues [47] found that scores over 145 discriminated between autistic people and people with other diagnoses (but not autism), including obsessive compulsive disorder and social anxiety disorder (even though samples were very small with only 12 patients in each group). The Dutch AQ was found to have satisfactory internal consistency and test/retest reliability. Its factor structure was determined in a combined student and general sample and was in line with the intended structure of the original AQ. Five factors were proposed: “social skill”, “attention switching”, “communication”, “imagination” and “attention to detail”. The first four factors were highly correlated and formed one higher order factor for social interaction, whereas the “attention to detail” scale was weakly correlated with the other factors.

### 2.3. Procedure

All participants were invited to complete the anonymous online survey via the Qualtrics platform and gave informed digital consent. The aim of the project was not disclosed to participants to avoid response bias. The study was framed as an investigation into a ‘new personality questionnaire’. Several demographic variables were assessed (age, gender, living situation, education level and occupation). Then, the original 50 AQ items followed by the AWE’s 52 items were presented and responses were forced (so that participants could not leave items missing). The survey ended with questions about the participants’ history of medical and psychological diagnoses and current medication, if applicable. Once the survey was complete, participants were debriefed and informed of the true purpose of the study.

### 2.4. Statistical Analyses

Using SPSS 26, two approaches for item reduction were employed to shorten the list of 102 items while retaining as much information as possible. Since the population of interest for this instrument concerned women, item reduction was conducted on women only (*n* = 331). First, we examined item discrimination by means of *t*-tests and mean difference scores between autistic and non-autistic women. We retained only items that were significantly different between groups (to adjust for multiple comparisons, the alpha criterion was divided by the number of comparisons and was set to < 0.001). Exploratory factor analysis (EFA) was conducted selecting the primary axis factoring (PAF) estimator, which is robust to non-normal distributions of ordinal data [65]. Items with low primary loadings (<0.35) and high cross-loadings (i.e., ratio of loadings > 0.75) were removed from the scoring (these cut-offs are recommended for broader scales [64]).

Reliability and validity of the remaining AWE items were tested using the entire sample including autistic and non-autistic men and women (*n* = 642). Total and subscale scores of the AWE were calculated as the sum of the retained items. Cronbach’s alpha was calculated to determine the reliability of the scales. Descriptive statistics of the AQ and AWE were computed per group, and 2 × 2 ANOVA (group × gender) tests and independent samples *t*-tests were carried out to test for mean group differences. Cohen’s d was calculated as a measure for effect size. Intercorrelations were determined by calculating Pearson correlations between scale scores. To examine the sensitivity and specificity of the AQ and the AWE for predicting autism diagnoses, statistical comparisons of area under the curves (AUCs) were carried out. The corresponding receiver–operator characteristic curves (ROC curves) were examined, and Kolmogorov–Smirnoff (K-S) metrics were used to identify suitable cut-offs for the AQ and AWE. In addition, potential differences in scores and sensitivity/specificity relating to age of diagnosis were explored by means of correlations. To estimate test/retest reliability in the convenience sample (*N* = 70), intraclass correlation coefficients (ICCs) were computed between the test and the retest scale scores.

## 3. Results

### 3.1. AWE

To select the items that discriminated best between autistic and non-autistic women, all 102 items (50 items originating from the Dutch AQ and 52 new items) were subjected to a discriminant analysis. See Appendix A for all included and excluded items and Appendix B for the *t*-tests per item (Table A1). Since nearly all of the items significantly discriminated between autistic and non-autistic women, only the 75 items with the largest mean difference between groups were included for further analysis. This decision was made to produce a screening questionnaire that could be completed by the participants in a reasonable time. This step removed 27 items (original AQ items 1, 6, 12, 14, 18, 21, 24, 29, 30, 40, 48 and 49 and new items 3, 5, 10, 13, 14, 18, 22, 24, 25, 30, 38, 40, 43, 47 and 48). The comparisons revealed that new item 20 should not be reverse-scored. Accordingly, this item was not reverse-scored in subsequent analyses. 

In the next step, an EFA was performed on the remaining 75 items in the women only. We used primary axis factoring with a promax rotation since an inter-correlation between the factors was expected, as all of the factors were intended to measure “autism”. Based on the intended six subscales and an examination of a scree plot, a six-factor model was initially selected. However, as only two items loaded significantly (>0.35) on the sixth factor, we decided that a five-factor solution was more appropriate, since having few items loading on a factor will lead to underrepresentation in the final scale [64]. The resulting five-factor model of the AWE included a total of 49 items, including 22 items originating from the original AQ and 27 new items (see Table 2 for the retained items and domain loadings). Based on this EFA, we removed 16 original AQ items (items 2, 4, 5, 7, 10, 11, 16, 17, 25, 28, 32, 37, 38, 39, 43 and 50) and 10 new items (items 1, 7, 12, 15, 16, 21, 23, 41, 45 and 51). After inspecting the items of each factor, the EmFASiS team members agreed to the following scale names: “social functioning and communication” (19 items), “initiative and social motivation” (11 items), “social intuition” (9 items), “sensing boundaries” (5 items, containing items related to sensory processing) and “attention to detail” (5 items). The complete five-factor model explained roughly 50.8% of the variance. The respective factors explained 32.6% (“social functioning and communication”), 6.1% (“initiative and social motivation”), 4.6% (“social intuition”), 4% (“sensing boundaries”) and 3.5% (“attention to detail”) of the variance in the AWE scores. The Kaiser–Meyer–Olkin (KMO) measure of sampling accuracy was 0.932, indicating its suitability for the factor analysis. The AWE and the scoring template are provided in Appendix A, respectively.

### 3.2. Intercorrelations and Reliability

The intercorrelations between the AWE subscales are presented in Table 3. Large correlations (r > 0.5) were found between all subscales and the total AWE score, indicating that all scales contribute to the measurement of autistic traits. All subscale scores except for the “attention to detail” subscale had large intercorrelations. The “attention to detail” subscale had medium-sized correlations (r > 0.3) with the other subscales.

The Cronbach’s alpha indicated excellent internal consistency (>0.70) for both the 50-item AQ (α = 0.910) and the 49-item AWE (α = 0.946). The “social interaction and communication” scale also demonstrated excellent internal consistency (α = 0.919). The other scales had good internal consistency scores, which were as follows: “social motivation” (α = 0.868), “social intuition” (α = 0.812) and “sensing boundaries” (α = 0.824). However, the “attention to detail” subscale demonstrated a moderate level of internal consistency (α = 0.766). The test/retest reliability was assessed with an interval of approximately 8 weeks and appeared to be good (>0.750) using ICCs (r(70) = 0.818, *p* < 0.001) for the AQ as well as the AWE (r(70) = 0.764, *p* < 0.001). The test/retest reliabilities for the subscales were moderate–good for both scales (ranging between r = 0.607 and 0.785 for the AQ and between r = 0.678 and 0.766 for the AWE).

### 3.3. Group Comparisons

Table 4 presents the descriptive statistics of the AQ and AWE scores and the subscales per group, and Table 5 presents the effect sizes of the group comparisons and the statistical differences. As expected, autistic individuals scored significantly higher on both the AQ and AWE compared to individuals from the general population. This held for the overall sample and for each gender, i.e., autistic women had higher AQ and AWE scores than non-autistic women, and likewise in men (all with large effect sizes). This significant score difference between the groups was larger for women than for men on the AWE, as indicated by larger effect sizes (Table 5). An additional 2 × 2 ANOVA with group (people in the general population, people with autism) and gender (women, men) as factors indeed confirmed an interaction of group and gender with a medium effect size (F(1,612) = 4.15, *p* = 0.042, η^2^ = 0.007). This means that the AWE scores of autistic women deviate to a greater extent from women in the general population than AWE scores of autistic men from men from the general population. This was not the case for the original Dutch AQ scores, since no group ×gender interaction was present (F(1,612) = 2.44, *p* = 0.119, η^2^ = 0.004). On the original Dutch AQ, women in the general population scored lower than men in the general population, while autistic men and women did not differ in their scores. On the AWE, however, no gender differences were found, either for the general population group or for the autism group. 

Regarding the subscales of the AWE, autistic individuals, as expected, scored significantly higher on all subscales compared to participants from the general population with large effect sizes. This held for the overall sample and according to gender. The “social functioning and communication” and “attention to detail” subscale scores differed subtly between autistic women and men. Specifically, autistic women scored higher on the “social functioning and communication” subscale with a medium effect size, indicating they experienced more difficulties in social communication compared to men. Autistic men, on the other hand, showed higher scores on the “attention to detail” subscale with a small effect size, indicating more traits in this area compared to women. No significant differences were found on the other subscales when comparing autistic men and women. Even though the men and women from the general population group did not differ on the AWE total scale, they did differ on all subscales with small-to-medium effect sizes. Men from the general population obtained higher scores than women from the general population on the following subscales: “initiative and social motivation”, “social intuition” and “attention to detail”. On the other hand, women obtained higher scores than men from the general population on the “social functioning and communication” and “sensing boundaries” subscales.

### 3.4. ROC Analyses

ROC analyses were used to assess the sensitivity and specificity of both the AQ and the AWE; the curves are presented in Figure 1. In the overall sample, both the AQ and AWE produced similar AUCs (AWE: AUC = 0.925; AQ: AUC = 0.922), which were not statistically different (Z = 3.26, *p* = 0.745), indicating that sensitivity and specificity were highly similar between the questionnaires. For women only, the AUCs were slightly higher (AWE: AUC = 0.936; AQ: AUC = 0.925), but the difference between the AWE and the AQ was not statistically significant (Z = 1.08, *p* = 0.278). For men only, the AUCs were slightly lower (AWE: AUC = 0.907; AQ: AUC = 0.917), and the AWE and AQ AUCs did not differ either (Z = −0.82, *p* = 0.411). This indicates that for men and women separately, the sensitivity and specificity of the AQ and AWE were highly similar.

Based on K–S metrics, the most optimal cut-off (with a K–S max = 0.73) for the AWE was 123+ (on a scale ranging from 49 to 196). At this cut-off score, 88% of autistic individuals were correctly classified as being autistic and 15% of people from the general population were misclassified as being autistic. There was no need for separate cut-offs for men and women, since the AUCs were approximately equal between the genders.

The classification statistics of the AQ were highly similar to the AWE. In our samples, the most optimal cut-off (with a K–S max = 0.72) for the AQ was 121+ (on a scale ranging from 50 to 200). At this cut-off score, 88% of autistic individuals were correctly classified and 16% of people from the general population were misclassified as having elevated autistic traits.

### 3.5. Early Versus Late Diagnosis

Autistic women and men did not differ in their self-reported age of diagnosis (F(1,133) = 1.26, *p* = 0.263) with a mean age of diagnosis of 31 years (SD 16) and a median of 30 years (0–63 years). Note that the participants were asked to estimate the age of diagnosis if they were not entirely sure, which resulted in a few people setting the age at an unrealistically young age, e.g., below the age of 2 years. Figure 2 presents the age distribution of the sample, showing a wide range, which allows for additional correlational analyses. Age of diagnosis correlated positively with both the total AWE (r(117) = 0.194, *p* = 0.034) and total AQ (r(117) = 0.251, *p* = 0.006) scores, indicating more autistic characteristics at a later age of diagnosis. Age of diagnosis only correlated positively with the sensing boundaries subscale (r(117) = 0.218, *p* = 0.018), indicating more difficulties on this scale amongst participants with a later age of diagnosis. Age of diagnosis did not correlate significantly with the other subscales of the AWE. However, when controlling for current age, which correlated strongly with age of diagnosis (r(117) = 0.910, *p* < 0.001), these significant correlations disappeared. Splitting the analyses according to gender did not change the above results. Current age also correlated positively with the sensing boundaries subscale for both women (r(79) = 0.230, *p* = 0.041) and men (r(59) = 0.280, *p* = 0.032), indicating that older individuals reported more difficulties on this subscale. Only for men, current age also correlated positively with the attention to detail subscale (r(59) = 0.347, *p* = 0.012), indicating that older men reported more difficulties on this subscale.

Exploring sensitivity and specificity related to age of diagnosis did not reveal any meaningful differences between the AQ and the AWE. In individuals who received their diagnosis at the age of 21 or later, the AQ had an AUC of 0.90 and the AWE had an AUC of 0.897 (Z = 0.186, *p* = 0.85). In people who received their diagnosis earlier in life, the AQ had an AUC of 0.938, while the AWE had an AUC of 0.945 (Z = 0.849, *p* = 0.396).

## 4. Discussion

### 4.1. Summary of the Main Findings

In this study, we developed the Experience of Autistic Women (the AWE) and tested its reliability and validity. The AWE provides a set of 49 items identifying the challenges that autistic women frequently report and matter most to them. We compared this with the AQ, which has demonstrated modest to good levels of reliability and strong predictive validity over the past twenty years. Autistic men and women achieved similar scores on both the AQ and the AWE, and both the AQ and AWE had equal predictive value for detecting the diagnosis of autism in both genders. In the ROC analyses, the predictive value of both the AQ and AWE was excellent (AQ: AUC = 0.922; AWE: AUC = 0.925), and no statistical differences were detected between men and women for these values. Whilst there was a gender difference on the AQ total score among non-autistic people (men > women), there was no gender difference among autistic people, confirming earlier studies [56]. On the AWE total score, no gender differences were found, either among autistic people or in the general population (differences in the subscale scores are discussed in Section 4.2). Since the predictive validity of both the AQ and AWE appeared excellent in both men and women, both instruments can be used to screen for autism irrespective of gender. The initial validation of the AWE was performed only on women since the study’s aim was to find autistic items that would be sensitive and specific to autistic women. However, the resulting AWE item list surprisingly applied equally well to autistic men, questioning the female-specificity of the AWE (which is further discussed in Section 4.2). Therefore, further validations of the AWE including all gender groups are warranted (see Section 4.5 for recommendations).

The original AQ can still be used as a primary screening instrument for autism in both men and women. However, while maintaining predictive validity, the AWE covers autistic characteristics that resonate with autistic women, which was ensured by co-creating our research and the items with experts by experience. The AWE adds relevant information about camouflaging behaviours, social initiative and motivation, and difficulties sensing boundaries which is not integrated into the original AQ. The AWE therefore improves the content validity (i.e., making it more representative of contemporary perceptions of autism) and also fulfils a pressing clinical need for autism measures that better resonate with autistic women. Recognizing themselves in the additional items may foster the self-acceptance of autism in women, which is crucial for their mental wellbeing [14] and which may eventually lead to a healthier coping style (e.g., less camouflaging).

The reliability and validity of both the AQ and the AWE appeared to be excellent. The internal consistency was excellent, exceeding a Cronbach’s alpha of 0.90 for both the AQ and AWE. The internal consistency of the AQ exceeded those in previous reports of the Dutch AQ [47,49]. The test/retest reliability was good for both the AQ (r = 0.818) and AWE (r = 0.764), showing that both instruments are stable over time. Also, the subscales of both measures had moderate-to-good levels of test/retest reliability. The predictive validity of both the AQ and AWE was excellent, evidenced by the fact that autistic individuals scored on average approximately two standard deviations higher than individuals from the general population. There is evidence supporting the discriminative validity of the AWE: the AWE scores of autistic women were significantly higher than those of women from the general population, and this difference was greater when examining the AWE scores of autistic men compared to men from the general population (2 × 2 interaction of group × gender). Thus, there is a greater difference in autistic traits between autistic women compared to women in the general population than in men. This is in line with previous findings in which the AQ showed attenuated gender differences within autistic individuals [56]. Nevertheless, both the AQ and the AWE can be used irrespective of gender, as the ROC analysis showed high levels of sensitivity and specificity in both genders (all AUCs were >0.90), which did not differ significantly from each other. Because of this independence of gender, we hypothesize that both the AQ and AWE may also be used for assessments of transgender and gender-nonconforming people; however, this needs to be tested in further research. Furthermore, both the AQ and the AWE performed well in distinguishing autistic individuals from individuals from the general population regardless of whether or not the diagnosis was made early or late in life.

### 4.2. Gender-Specific Findings

It was surprising that the predictive value of the AQ and AWE was highly similar between the genders, since we expected that the AWE would be more predictive of autism in women and the AQ would be more predictive of autism in men. The AWE items were based on trait descriptions of autistic women, based on a thorough review of the latest literature, interviews with people diagnosed as autistic, and expert opinions [4,63]. The current study suggests that autistic men also experience the autistic characteristics that are reported in the literature to be female-specific. This is an interesting finding since it stresses that autistic men and women may be more similar than often thought, corroborating previous findings with the AQ-10 [66] and full-scale AQ [56] as well as recent qualitative findings [63]. However, subtle gender differences were found on the subscale profiles of the AWE (with small-to-medium effect sizes), contrasting the strong effects of the groups (autism versus general population—with large effect sizes). The differences in autistic traits between autistic men and women in our sample are therefore rather subtle.

These subtle gender difference in autistic individuals presented in the domains of “social interaction and communication” (more challenging for autistic women) and “attention to detail” (more challenging for autistic men), but this pattern of gender differences was also seen in the general population sample (which is discussed below). Autistic women expressed significantly more “social interaction and communication” difficulties than autistic men. Compared to the original AQ items about difficulties with conversations, social situations and friendships, this new scale contains many items related to camouflaging these difficulties. These difficulties are reflected in items such as “I often think out a conversation in advance”, “I usually adapt to the other person in a friendship” and “I go to social events when I don’t want to because it’s expected of me”. This is in line with previous findings that autistic women use more strategies to appear less autistic [12,13,32,33,67]. Compared to the general population, autistic women deviated more from the female norm than autistic men do from the male norm, which may indicate that there is more pressure on autistic women to act neurotypically [68]. This is not surprising, as girls and boys have already been shown to have different (same-gender) peer relationship styles, a difference that increases with age [69]. Compared to boys, girls engage in more prosocial interactions, are more focused on connections and more sensitive and reactive to distress and emotion in others. This fosters tighter and more intimate social networks, but also may heighten vulnerability to emotional difficulties. The pressure on autistic girls and women to be accepted and seen as typical may therefore be greater than in boys and men, but this causal explanation needs to be further investigated in autistic adults.

Autistic men, on the other hand, scored significantly higher on the “attention to detail” subscale compared to autistic women. The items in this scale remained very similar to the original AQ on the “attention to detail” subscale; in the AWE version, more examples of interests are given, so that both men and women could identify with them. This is reflected in items such as “I like to collect information about categories of things (e.g., types of car, types of bird, types of train, types of plant, etc.)” and “I notice patterns in things all the time”. This finding is in line with previous studies showing a more prominent presence of restricted, repetitive patterns of behaviour, interests or activities (RRBIs) in males compared to females [39]. Greater attention to detail scores in autistic men might reflect the higher prominence of rather uncommon fascinations/interests [40], whereas autistic women have fascinations/interests that are more socially accepted or stimulated by their environment, and therefore are not necessarily considered as autism-specific fascinations/interests [3]. Although autistic men present with a stronger attention to detail, autistic women again differed more from the neurotypical norm than men on this subscale. 

The scores of autistic men and women on difficulties with “initiative and social motivation”, “social intuition” and “sensing boundaries” were strikingly similar. This pattern differed from that of people in the general population, who show subtle gender differences with small-to-medium effect sizes on all subscales of the AWE. Women from the general population score higher on the “social functioning and communication” and “sensing boundaries” subscales, whereas men from the general population score higher on the subscales: “initiative and social motivation”, “social intuition” and “attention to detail”. Autistic traits are therefore present in the general population, but the type of traits differ according to gender. 

### 4.3. Sensing Boundaries

In accordance with the DSM-5, we aimed for a new subscale to be added to the AWE that would include sensory processing difficulties. However, many of the sensory items were removed after the discrimination and factor analysis of the items. Difficulties with sensing the need for food and drink did not discriminate between the general population and autistic women, indicating that these sensory problems are also highly common in the general population. Difficulties with tiredness and being in pain did not fit the factorial structure of the AWE, indicating that these physical complaints do not contribute to the hierarchical structure of autism. One of the anticipated sensory (reverse-scored) items “I feel comfortable in a busy environment with a lot of noise, light, and/or smell” fitted into the “social functioning and communication scale”. This indicates that the avoidance of highly stimulating environments strongly relates to being in social situations. Our recent qualitative study explains this finding, pointing out that autistic individuals attributed their tiredness after social engagement to the sensory stimulation more so than to the social interaction itself [63]. The sensory processing items that were included in the factor structure formed a new scale, which we labelled as “sensing boundaries”. This included items such as “When I’ve done too much, I only feel it afterwards” and “I’ve been doing more than I can actually handle all my life”. “Sensing boundaries” pertains to sensing your individual needs, managing your own energy and resting when needed. The fact that this scale fits into a measure of autistic traits (with medium intercorrelations), shows that sensing boundaries is a specific skill that poses difficulties for autistic individuals, both in men and in women. This skill potentially relates to confusion about interoceptive bodily states (alexithymia), which has recently been found to be present in as many as 74% autistic individuals [70]. Particular difficulties regarding the interoception of autistic individuals pertained to “(a) challenges with identifying and describing bodily states (b) not feeling the affective/motivational components to act upon bodily states and (c) externally cued physiological self-regulation” (p. 3363, [70]). Difficulties with interoception and sensing boundaries may make autistic people prone to developing “autistic burnout”, which is a common phenomenon in autism that occurs when the expectations of the environment or themselves exceed the person’s skills, resulting in a state of chronic exhaustion, a loss of skills and a reduced tolerance to stimuli [33]. 

Interestingly, particularly the “sensing boundaries” subscale score was increased in autistic individuals who received a late diagnosis (which were also the older individuals in our sample), indicating more difficulties in sensing individual needs, managing one’s own energy and resting when needed. One explanation could be that individuals diagnosed later in life, as opposed to individuals diagnosed early, developed fewer coping skills because they could not profit from professional care early in life [71]. Also, individuals diagnosed later in life run the risk of masking of their autism and pretending to be ‘normal’ for a longer time, increasing the risk of systematically crossing their own (energy) boundaries. A second explanation for more difficulties with sensing boundaries in autistic individuals diagnosed later in life (and older individuals) is that interoceptive abilities are known to be affected by the ordinary aging process. Aging in the general population relates to lower interoceptive awareness and declining interoceptive accuracy [72], which may also be true for the autistic population. A final, sociological explanation is that with increasing age, new and additional social roles are fulfilled, such as becoming a romantic partner, a parent and/or an employee, which require multitasking and complicate one’s social life. More complex social lives may make it more difficult to sense and protect one’s own boundaries, particularly when this skill is less well developed. Interestingly, women in the general population reported more problems than men on the sensing boundaries scale, which could also be related to the more complex social lives of women compared to men, with more social roles to fulfil (e.g., combining childcare with employment). Future works could investigate how these gender differences relate to differences in (beliefs about) gender and social roles [73].

### 4.4. Clinical Recommendations

In this study, both the original AQ and the AWE were shown to be excellent measures of autistic traits in men and women with good levels of reliability and validity. The AWE adds clinical value by providing updated subscales that autistic women can relate to, which are sensitive to subtle gender differences in autistic individuals. Five new subscales are proposed: “social functioning and communication”, “initiative and social motivation”, “social intuition”, “sensing boundaries” and “attention to detail”. Compared to the AQ scales, these scales add additional information about camouflaging behaviours, social initiative and motivation, and difficulties with sensing boundaries. The AWE therefore contributes to gender-sensitive care by acknowledging autistic characteristics that are more commonly observed in women. The English AWE and the scoring template are freely available in Appendix A and the Dutch and English version can be freely downloaded from emfasisonderzoek.org (accessed on 4 December 2023) for use in clinical practice and research. Based on our sensitivity and specificity analyses, we recommend using a cut-off of 123+ for the AWE and 121+ for the AQ (when using all Likert scale points 1–4). The false positive rates are ~15% at these cut-offs. Given that AWE could be used as a screening instrument that may call for a more thorough diagnostic process in mental health care services, a substantial number of false positives is acceptable. This AQ cut-off lies in between the proposed cut-off for ‘subthreshold autistic traits’ (114+) and ‘autism according to DSM-IV’ (145+) when using the Dutch AQ [47]. The suggested cut-offs in the present study are applicable to the entire autism spectrum, in accordance with the DSM-5. Gender appeared not to affect the threshold on the AQ or AWE, which may suggest a promising insight for the screening of transgender and gender-nonconforming people, even though this needs to be validated in future research. This is especially relevant in autism, since different sexual orientations and gender diversity are more common in individuals with autism than in the general population, particularly in autistic women [74].

Even though clinical settings need short screening questionnaires to identify the right group of people to be assessed for an autism diagnosis, long forms are still useful (the AQ contains 50 items and the AWE contains 49 items). Particularly, the subscale profile of the AWE provides clinically relevant gender-sensitive information to the challenges autistic individuals are facing, which can be used in psychoeducation and the selection of health care services. While a short version of the AQ [75,76,77] is often used to aid the referral process in primary care settings (for example by general practitioners), these are often limited in terms of internal consistency (and therefore reliability) [26]. Furthermore, they do not capture the full range of autistic characteristics, which may subtly differ between genders (see Section 4.2). Lastly, informants other than the individual with suspected autism are necessary in the screening process, because a general limitation of self-report questionnaires is that they rely on self-reflection and self-awareness, which can be reduced in autistic individuals [78]. Therefore, AQ and AWE scores may over- or underestimate the true autistic traits of participants. 

During the process of this research study, another new screening questionnaire for adult autistic women was proposed: the Girls Questionnaire for Autism Spectrum Condition (GQ-ASC) [79]. The GC-ASC is an adult version of a self-report screening questionnaire for adolescent girls, which includes adult as well as retrospective childhood items. Its content is partially overlapping with the AWE (e.g., camouflaging, socializing difficulties and feminine interests) and partially diverting from it (e.g., it includes a scale for imagination and play items as well as sensory sensitivities which are not part of the AWE). The clinical advantage of the AWE over the GC-ASC is that all items were drawn from the adult presentation in cooperation with experts by experience. The GC-ASC may provide more information about childhood symptoms, which are not taken into account in the AWE.

### 4.5. Strengths and Limitations

The strengths of the current study are that the AWE items were developed in accordance with guidelines for scale development [64]. We wrote the items based on a thorough review of the recent literature, interviews with autistic people, expert opinions as well as co-creation with experts by experience. Furthermore, we validated the items in a large sample of people diagnosed with autism (*N* = 138) with an equal representation of women and men. Both the AWE and AQ appeared to have excellent psychometric qualities and are free to use.

Several limitations apply to this study. Firstly, since men as well as transgender and gender-nonconforming people were not included in the discriminant and factorial analysis, the AWE may not have the perfect fit for (autistic) men or gender-diverse people. Even though recent new insights in autistic traits have mainly involved women, it is necessary to replicate the findings in men and women, as well as groups of transgender and gender-nonconforming people. In the future, a revised contemporary version of the AQ may be developed which is suited for all gender groups and which provides gendered norm data. Secondly, there are several statistical limitations to our study. The sample size of *n* = 331 was relatively low for an EFA with 75 items and five factors [80]. Larger samples are necessary to replicate the suggested factor structure (including a parallel analysis for categorical data in addition to a scree plot) and to conduct a confirmatory factorial analysis in an independent sample. Particularly, a stratified confirmation of the factor structure for men, women and gender-diverse people is warranted, since the invariance of the construct is a prerequisite to reliably comparing group means [81]. The current initial validation of the AWE did not delve into gender-related measurement invariance, i.e., the differential endorsement of items by men and women, which could be detected by using the item response theory. It is recommended that a replication study makes use of the item response theory for the detection of gender bias on the item level. A third limitation concerns the representativeness of the sample; the autistic participants were recruited from a tertiary care institution offering specialized care for autism, and therefore the psychometric properties may not be as good for autistic individuals who remain undiagnosed or who are not receiving specialized care. Additionally, we did not test for the differential diagnostic properties of the AWE. It is therefore unclear whether the AWE would also be sensitive to other psychiatric diagnoses, such as anxiety, mood, substance use or personality disorders. Co-occurring mental health conditions in autism are very common [10], so it will be difficult to assess the specificity of the AWE in a group of autistic people without these conditions (more than half of the current autism sample reported secondary psychiatric diagnoses). However, previous work has indicated that the Dutch AQ discriminates well between autism and social anxiety disorder as well as obsessive compulsive disorder [47], but only small, carefully selected samples of young people participated in that study (*n* = 12 in each group). Generally, in mental health care settings, the sensitivity and specificity of the AQ appeared lower in comparison to those of studies using samples from the general population as a control group [49,82,83]. To prevent the overdiagnosis of autism and long waiting lists for a specialized assessment, it is necessary to further investigate differential diagnostic qualities of the (Dutch) AQ and AWE.

## 5. Conclusions

We conclude that the AWE offers a valuable additional tool to identify the challenges that autistic women frequently report. We recommend that the original AQ is used as a primary screening instrument for autism in both men and women. The AWE can additionally be used in clinical and research settings to cover all relevant aspects of autism in women. Future works, in close collaboration with the authors of the original AQ, may consider revising the original AQ into a contemporary version with an improved fit for autistic men, autistic women and gender-diverse people. Other female-specific questionnaires for autism (the GQ-ASC) may also be incorporated in this validation process. The AWE brings us one step closer to gender-sensitive care in the field of autism.

## Figures and Tables

**Figure 1 ijerph-20-07148-f001:**
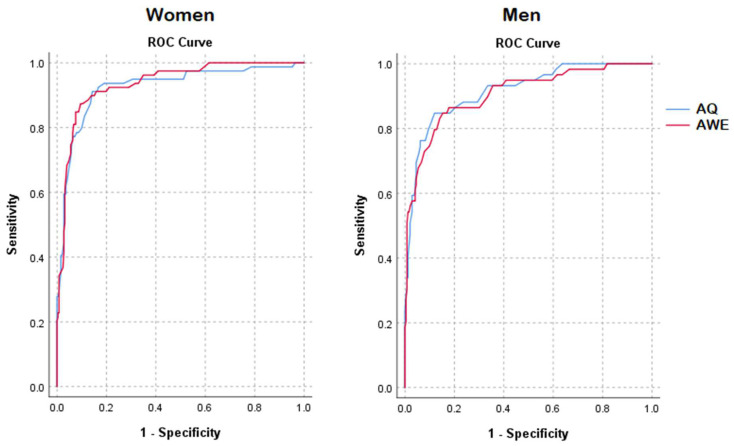
ROC curves discriminating individuals with ASD from the general population via the AWE and AQ and separated by gender.

**Figure 2 ijerph-20-07148-f002:**
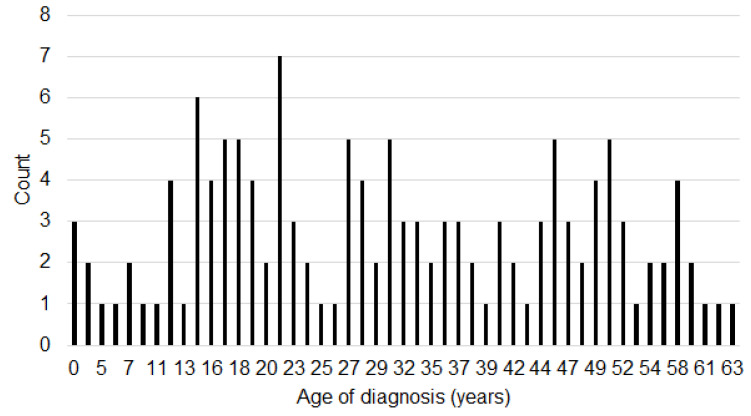
Histogram displaying the age of diagnosis.

**Table 1 ijerph-20-07148-t001:** Demographic characteristics.

	Women		Men	
	GP (*n* = 238)	ASD (*n* = 79)	GP (*n* = 236)	ASD (*n* = 59)
Age	37 (SD 15)	36 (SD 14)	49 (SD 20)	42 (SD 14)
Age of diagnosis (mean in years)	-	33 (SD 18)	-	30 (SD 14)
Age of diagnosis (min–max)		6–59		4–63
Diagnosed before/after age 18 years		16/55 (8 missing)		14/40 (5 missing)
Living situation				
Single	63 (26.5%)	30 (38.0%)	59 (25.0%)	36 (61.0%)
Living together	151 (63.4%)	29 (36.7%)	161 (68.2%)	17 (28.8%)
Assisted living	1 (0.4%)	4 (5.1%)	1 (0.4%)	2 (3.4%)
Different	23 (9.7%)	16 (20.7%)	15 (6.4%)	3 (5.1%)
Highest level of education attained				
Elementary education	0	2 (2.5%)	3 (1.3%)	1 (1.7%)
Elementary education + other than the following	2 (0.8%)	0	2 (0.8%)	1 (1.7%)
Junior secondary vocational/general education	7 (2.9%)	7 (8.9%)	15 (6.4%)	5 (8.5%)
Senior secondary vocational/general education or pre-university education	49 (20.6%)	23 (29.1%)	63 (26.7%)	18 (30.5%)
Higher professional education (BSc)	121 (50.8%)	39 (49.4%)	116 (49.2%)	19 (32.2%)
Academic education (MSc)	55 (23.1%)	5 (6.3%)	36 (15.3%)	6 (10.2%)
Other	4 (1.7%)	3 (3.8%)	1 (0.4%)	8 (13.6%)
Educational program				
No	164 (68.9%)	64 (81.0%)	188 (79.7%)	55 (93.2%)
Full-time	54 (22.7%)	14 (17.7%)	37 (15.7%)	2 (3.4%)
Part-time	20 (8.4%)	1 (1.3%)	11 (4.7%)	1 (1.7%)
Job status				
No	38 (16.0%)	45 (57.0%)	96 (40.7%)	38 (64.4%)
Full-time	48 (20.2%)	8 (10.1%)	83 (35.2%)	11 (18.6%)
Part-time	152 (63.9%)	26 (32.9%)	57 (24.2%)	9 (15.3%)

**Table 2 ijerph-20-07148-t002:** Item numbers, item content and loadings on the five domains of the AWE scoring template, ordered per domain loading.

Item Numbers ^a^	Domain Labels and Items ^b^	Domain Loadings
	Social functioning and communication (SFC)	
NEW42	I often think out a social event completely in advance *	0.819
NEW39	During a conversation, I am continuously consciously observing how the other person reacts to me *	0.787
NEW27	I don’t have to think about how I’m going to act toward others	0.767
NEW49	I usually consciously watch others to see what I should say *	0.724
NEW29	I get upset when someone else’s mood isn’t good *	0.702
NEW50	I often think out a conversation in advance *	0.672
NEW44	I become quiet in a conversation with several people *	0.602
NEW28	I go to social events when I don’t want to because I’m afraid I won’t belong otherwise *	0.592
NEW31	After a conversation, I often think for a long time about what someone actually meant *	0.573
AQ46	New situations make me anxious *	0.554
NEW8	I usually adapt to the other person in a friendship *	0.542
AQ26	I frequently find that I don’t know how to keep a conversation going *	0.534
NEW34	I automatically know the best thing to say in a conversation	0.533
NEW46	I am satisfied with my relationships with my friends, family, or partner even if they are not perfect	0.507
NEW2	I go to social events when I don’t want to because it’s expected of me *	0.505
AQ33	When I talk on the phone, I’m not sure when it’s my turn to speak *	0.482
NEW19	I feel that I really fit in during a conversation	0.478
AQ22	I find it hard to make new friends *	0.443
NEW17	I am good at keeping up friendships	0.386
	Initiative and social motivation (ISM)	
AQ44	I enjoy social occasions	0.789
NEW4	Contact with other people gives me energy	0.781
AQ47	I enjoy meeting new people	0.768
NEW35	People and their behaviour don’t interest me much *	0.583
AQ13	I would rather go to a library than a party *	0.564
AQ15	I find myself drawn more strongly to people than to things	0.550
NEW32	I usually feel a connection quickly when I’m talking to others	0.539
NEW20	After a few days of doing nothing, I really have to do something	0.483
AQ34	I enjoy doing things spontaneously	0.473
NEW37	I feel comfortable in a busy environment with a lot of noise, light, and/or smell	0.457
NEW33	I’ve been pretending to be social for years, but I’m actually not *	0.395
	Social Intuition (SI)	
AQ36	I find it easy to work out what someone is thinking or feeling just by looking at their face	0.646
AQ27	I find it easy to “read between the lines” when someone is talking to me	0.581
AQ45	I find it difficult to work out people’s intentions *	0.540
AQ31	I know how to tell if someone listening to me is getting bored	0.525
AQ3	If I try to imagine something, I find it very easy to create a picture in my mind	0.521
AQ20	When I’m reading a story, I find it difficult to work out the characters’ intentions *	0.510
AQ8	When I’m reading a story, I can easily imagine what the characters might look like	0.492
AQ42	I find it difficult to imagine what it would be like to be someone else *	0.487
AQ35	I am often the last to understand the point of a joke *	0.480
	Sensing boundaries (SB)	
NEW11	I am in touch with my body and rest when I need to	0.717
NEW26	I notice when I am exhausted or overloaded in time	0.652
NEW52	When I’ve done too much, I only feel it afterwards *	0.519
NEW36	I’ve been doing more than I can actually handle all my life *	0.431
NEW9	I have enough energy to do normal daily activities	0.411
	Attention to detail (AD)	
AQ9	I am fascinated by dates *	0.742
AQ19	I am fascinated by numbers *	0.719
AQ41	I like to collect information about categories of things (e.g., types of car, types of bird, types of train, types of plant, etc.) *	0.448
AQ23	I notice patterns in things all the time *	0.424
NEW6	I know everything about one subject, for example about animals, diets, historical periods or upbringing *	0.376

^a^ = item numbering follows the long forms of the AQ and new items; ^b^ = disagreement to the items is considered an autistic response style (except for reverse-keyed items); * = items are reverse-scored.

**Table 3 ijerph-20-07148-t003:** Intercorrelations of the AWE scales.

	1	2	3	4	5	6
1. AWE	-	0.920 **	0.835 **	0.714 **	0.743 **	0.557 **
2. SFC		-	0.683 **	0.522 **	0.662 **	0.399 **
3. ISM			-	0.548 **	0.536 **	0.345 **
4. SI				-	0.453 **	0.339 **
5. SB					-	0.299 **
6. AD						-

SFC = social functioning and communication; ISM = initiative and social motivation; SI = social intuition; SB = sensing boundaries; AD = attention to detail. ** *p* < 0.01.

**Table 4 ijerph-20-07148-t004:** Means and standard deviations of the AQ and AWE scores separated by gender and ASD diagnosis.

	Women		Men	
	NT (*n* = 238)	ASD (*n* = 79)	NT (*n* = 236)	ASD (*n* = 59)
AQ	104.6 (14.5)	139.0 (17.6)	109.0 (13.5)	138.9 (16.1)
AWE	100.7 (18.6)	143.7 (17.1)	102.4 (16.9)	138.3 (19.5)
SFC	41.7 (8.0)	58.7 (8.3)	39.7 (8.4)	53.5 (9.4)
ISM	20.4 (6.2)	30.3 (6.1)	21.5 (5.7)	29.5 (6.6)
SI	16.0 (4.3)	22.9 (5.0)	17.3 (3.9)	22.9 (4.7)
SB	11.5 (3.4)	16.3 (2.6)	10.2 (2.7)	15.5 (3.0)
AD	11.2 (4.0)	15.6 (3.6)	13.7 (3.4)	16.9 (3.8)

GP = general population; ASD = Autism Spectrum Disorder; SFC = social functioning and communication; ISM = initiative and social motivation; SI = social intuition; SB = sensing boundaries; AD = attention to detail.

**Table 5 ijerph-20-07148-t005:** Effect sizes (Cohen’s d) and statistical significance of group differences.

	GP Women vs. GP Men	ASD Women vs. ASD Men	ASD Women vs. GP Women	ASD Men vs. GP Men
AQ	−0.31 ***	0.01 ^ns^	2.13 ***	2.01 ***
AWE	0.26 ^ns^	0.29 ^ns^	2.41 ***	1.97 ***
SFC	0.24 *	0.59 ***	2.09 ***	1.55 ***
ISM	−0.18 *	0.13 ^ns^	1.61 ***	1.3 ***
SI	−0.32 ***	0.00 ^ns^	1.48 ***	1.3 ***
SB	0.42 ***	0.28 ^ns^	1.26 ***	1.86 ***
AD	−0.67 ***	−0.35 *	1.16 ***	0.89 ***

GP = general population; ASD = Autism Spectrum Disorder; SFC = social functioning and communication; ISM = initiative and social motivation; SI = social intuition; SB = sensing boundaries; AD = attention to detail. *** *p* < 0.001; * *p* < 0.05; ^ns^ = not significant.

## Data Availability

The data presented in this study are available on request from the corresponding or last author. The data are not publicly available due to the inclusion of privacy-sensitive data.

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
