# Peer review of "Measuring the Autistic Women’s Experience (AWE)"

_ijerph, 2023, doi:10.3390/ijerph20247148_

Round 1
Reviewer 1 Report
Comments and Suggestions for Authors
The title is clear and concise, reflecting the focus of the study. The abstract provides a comprehensive summary of the research, including the purpose, methodology, key findings, and implications. It effectively highlights the development of the AWE and its comparison with the established AQ. The language is precise and the results are well-documented. However, it could benefit from a brief statement about the limitations of the study to provide a more balanced view.
Introduction:
The introduction of the article provides a thorough overview of the research problem concerning the clinical diagnosis of autism in females and its associated challenges. It emphasizes the delayed or misdiagnosed nature of autism in females compared to males, highlighting the negative outcomes such as reduced societal participation, impaired quality of life, and mental health issues. The introduction presents a compelling argument for the need to address the gender differences in autism and underscores the importance of developing a more gender-sensitive diagnostic instrument.
The section presents a comprehensive background, discusses the historical neglect of gender differences in autism research, and notes the evolution of the male-to-female gender ratio in autism diagnoses. It effectively describes the gender-related differences in autism characteristics, notably the subtlety of symptoms in autistic women, their social interaction challenges, and their coping mechanisms, such as camouflaging.
The introduction convincingly sets the stage for the research and highlights the limitations of current diagnostic instruments, particularly the Autism-Spectrum Quotient (AQ), for assessing autistic traits in women. The need for a more comprehensive measure, the Autistic Women’s Experience (AWE), is well-justified.
Suggestions for improvement:
While the introduction is comprehensive, it is quite lengthy and could be condensed to maintain reader engagement. Consider breaking it into smaller sections to enhance readability.
The introduction could benefit from a concise summary of the aims and objectives of the study to provide a more structured preview of the research to follow.
It would be helpful to include a brief section on the importance of the current research, emphasizing how it addresses the gaps in existing diagnostic tools and sets the stage for the subsequent sections.
Incorporate a clear, concise statement of the research question or hypothesis at the end of the introduction to provide a precise focus for the study.
Materials and Methods:
The Materials and Methods section provides a detailed description of the study's participants, data collection, instruments, and statistical analyses. Overall, the section is well-structured and contains valuable information. Here are some points for consideration:
Clarity and organization: The section is well-organized and logically presented. The subsections, such as "Participants," "Materials," and "Procedure," make it easy for the reader to follow the research process.
Participant recruitment: The description of participant recruitment is thorough and clear. It includes information about the source of participants for both the autism group and the general population sample. The response rates, inclusion criteria, and reasons for exclusion are well-documented, providing transparency and helping to ensure the study's validity.
Demographic characteristics: The inclusion of demographic characteristics in Appendix A is useful. However, consider presenting key demographic information in the main text to provide readers with a clear overview of the participants.
Materials: The section effectively describes the development of the Autistic Women’s Experience (AWE) questionnaire and its items. The process of drafting items, conducting interviews, and using the Delphi consensus study is well-documented. Additionally, the section explains how the AWE differs from the Autism Spectrum Quotient (AQ) and why it was developed.
Procedure: The procedure for data collection is clearly outlined, from informed consent to debriefing participants. This transparency is crucial for understanding the ethical aspects of the study. However, the description of the procedure could be more concise while still retaining essential information.
Statistical analyses: The description of statistical analyses is comprehensive. It provides a clear understanding of the item reduction process for the AWE and the validity testing. Mentioning the statistical software used (SPSS 26) is essential for transparency.
Validation of AWE items: While the section mentions that item reduction was based on significant differences between autistic and non-autistic women, it could benefit from further explanation. Discuss the rationale behind setting the significance level at <0.001 and the implications for item selection.
Clarity of results: Consider improving the clarity of results. Mention if any specific items were excluded based on the item reduction process. Additionally, when discussing results, provide a brief summary of the major findings before diving into the statistical details.
Consider using tables or figures: To enhance the presentation of results, consider using tables or figures to display key statistics and results.
In summary, the Materials and Methods section is comprehensive and well-structured. However, it can be refined for conciseness in certain areas while maintaining clarity and transparency. Additionally, a brief summary of the major findings can aid readers in understanding the subsequent sections.
Results:
Explanation of Methods:
The text provides a clear description of the methods employed, including statistical analyses (e.g., discriminant analysis, EFA, ROC analysis).
The authors could improve the clarity by briefly explaining some technical terms. For example, explaining what "EFA" stands for (Exploratory Factor Analysis) might benefit readers who are not familiar with the abbreviation.
Statistical Details:
The text includes essential statistical information such as Cronbach's Alpha, correlations, effect sizes (Cohen's d), and ROC analysis. However, adding a brief interpretation of what these statistics mean would be beneficial for readers who may not have an in-depth understanding of statistics.
Repetition and Conciseness:
There is some repetition, especially when discussing the results. The text mentions the same findings (e.g., AUC values) multiple times in different contexts. To improve clarity and conciseness, these findings could be presented once, and the relevance to subsequent sections should be made clear.
Verb Tense:
Ensure consistent use of verb tense. In scientific writing, the past tense is often used for describing methods and reporting results, while the present tense is used for general facts or interpretations.
The discussion in this scientific article is an integral part of the paper, providing insights into the study's findings and their implications. Here's a critical evaluation of the discussion section:
Strengths of the Discussion:
Clear Summary of Findings: The discussion effectively summarizes the main findings of the study, including the development and validation of the AWE and its comparison with the AQ.
Discussion of Gender and Diagnostic Equality: The article rightly emphasizes that both the AQ and AWE demonstrated strong psychometric properties and are equally valid for diagnosing autism in both men and women. This is an essential point as it confirms the utility of these instruments across genders.
Identification of Gender Differences: The discussion identifies subtle gender differences, particularly in domains like "social interaction and communication" and "attention to detail." This adds depth to the understanding of how autistic traits manifest differently in men and women.
Suggestion for Clinical Use: The discussion provides practical recommendations for clinical use, suggesting specific cutoff scores for both the AQ and AWE. This guidance can be beneficial for healthcare professionals.
Discussion of Limitations: The authors acknowledge several limitations, such as the need for further analysis across genders and in undiagnosed populations, as well as the potential impact of co-occurring mental health conditions. This demonstrates a commitment to transparency and a cautious interpretation of the results.
Areas for Improvement:
Lack of Further Exploration: While the discussion acknowledges the need for additional analysis across genders and in other populations, it might benefit from proposing specific directions for future research. What kind of studies are needed, and what questions should be addressed to further advance our understanding of autism in men and women?
Interpretation of Gender Differences: While the study identifies subtle gender differences in autistic traits, the discussion could delve deeper into the potential implications of these differences, especially in the context of clinical practice. What might these differences mean for the diagnosis and treatment of autism in men and women?
Consideration of Diversity: The study hints at the possibility of autism being related to different sexual orientations and gender diversity, particularly in autistic women. A more in-depth discussion of this topic and its relevance would be valuable.
Clarity of Language: There are some sentences that could be made more concise for clarity. For example, in certain parts, the text becomes slightly complex and might benefit from more straightforward language.
The concluding section of this scientific article provides a concise summary of the study's key findings and implications. Here's a critical evaluation of the conclusions:
Strengths of the Conclusions:
Clear and Concise: The conclusions are straightforward and concise, effectively summarizing the study's main outcomes and recommendations.
Recommendations: The authors offer practical recommendations based on their findings. They suggest that the original AQ remains a valid primary screening instrument for autism in both genders. They also emphasize the potential value of the AWE in clinical and research settings, particularly for its coverage of autism-related challenges in women.
Acknowledgment of Collaborative Future Work: The conclusion indicates a willingness to collaborate with the authors of the original AQ to improve its fit for both men and women, which demonstrates a commitment to advancing the field.
Areas for Improvement:
Future Work Elaboration: While the conclusion mentions the possibility of revising the AQ into a contemporary version, it could benefit from a more detailed description of what such a revision might entail. This might include specific recommendations for improvements and a discussion of the challenges involved.
Application to Clinical Practice: The conclusion mentions "gender-sensitive care," but it could provide more explicit guidance on how these instruments, the AQ and AWE, should be used in clinical practice to enhance the care of autistic individuals.
Implications for Policy: The conclusion could discuss the potential implications of their findings for autism-related policies, particularly in areas like healthcare access, insurance, and educational accommodations.
Incorporation of Limitations: While the conclusion acknowledges the limitations of the study, it could briefly discuss how these limitations might impact the practical use of the AQ and AWE.
Reviewer 2 Report
Comments and Suggestions for Authors
Thank you for the opportunity to review this interesting paper, which reports a new Dutch language measure of autistic traits relevant to women. I would offer the authors the following comments.
Introduction
1. The authors generally provide a very helpful background to their study in the Introduction.
2. Lines 148-151: “Several studies evaluating different versions of the AQ (AQ-short, adolescent AQ, adult AQ) suggest that they are valid and reliable measures of autistic traits, irrespective of gender (Baron-Cohen et al., 2001; Hoekstra et al., 2008; Wakabayashi et al., 2006).”
Lines 188-189: “Whilst results from the AQ have been widely replicated, including in big data, autistic women have reported that it may not fully capture their unique experience.”
It is important to highlight that there have been noted inconsistencies in the factor structure (structural validity) of the AQ in the literature. Studies by English et al. (2020), Belcher et al. (2023), Lau et al. (2013), Kloosterman et al. (2011), and Hoekstra et al. (2008) have reported relatively poor fit indices for the original five-factor model. Concerns regarding shorter versions have also been raised, as seen in studies such as Lau et al. (2013), Taylor et al. (2020) and English et al. (2020). Additionally, various studies have reported less than satisfactory internal consistency reliability (Cronbach’s alpha, α < 0.7) for each factor in the original AQ model, including Lau et al. (2013), Freeth et al. (2013), Hurst et al. (2007), and English et al. (2020). Given these inconsistencies, it may not be accurate to assert, without caveat, that there is robust evidence supporting the psychometric validity of the AQ and that the results have been widely replicated. It would also be beneficial to include a reference supporting the statement that autistic women have reported that AQ may not fully capture their experiences.
3. Lines 154-161: “We propose that an additional measure is needed that better reflects autistic women’s experience. We developed the Autistic Women’s Experience (AWE) and co-created its items with autistic women, since involvement of experts by experience is fruitful in the research process and in the development of questionnaires (Brett et al., 2014). The AWE was constructed to reflect autistic characteristics that matter to autistic women. The AWE is timely since the AQ was designed two decades ago, when the understanding of autism was predominantly male-focused. Very few autistic women were included in the first validation study (13 women vs 45 men; Baron-Cohen et al., 2001).”
Lines 172-178: “Even though the Camouflaging Autistic Traits Quotient (CAT-Q) (Hull, Mandy, et 172 al., 2019) and the Sensory and Perception Quotient (SPQ) (Tavassoli et al., 2014) were developed to cover these issues, the AWE attempts to capture the full range of characteristics that might be more common among autistic women.”
Considering that a measure had been developed for the assessment of autism in women - the modified Girls Questionnaire for Autism Spectrum Condition (Brown et al., 2020), it would be valuable to highlight the new contributions of the AWE in the context of the existing measure.
4. Lines 177-186: “To date, this potential diagnostic gender bias of the AQ has not received much attention. One study did use item response theory to analyse gender bias in the short version of the AQ (AQ-10) (Murray et al., 2017). This study found evidence of gender bias on specific items on the AQ-10, although these biases at times favoured men and at other times favoured women, cancelling out a gender bias in the overall total score. This finding corresponds with results from a big data study (including over 800 autistic men and women) 183 on gender differences on the total AQ score. This study reported small gender effects within the autism group, indicating that typical gender differences (typically of medium effect size) are attenuated in autistic men and women (Baron-Cohen et al., 2014).”
It may be beneficial to discuss the findings on differential item functioning of AQ-10 in relation to other studies on gender-related measurement invariance of different AQ models. Please see for instance Murray et al. (2019), Grove et al. (2017), Jia et al. (2019), English et al. (2020), and Belcher et al. (2023). Our understanding is that Baron-Cohen et al. (2014) assesses gender differences in AQ scores rather than gender-related bias in the measurement tool itself, which is the focus of Murray et al. (2017) paper.
Methods and Results
1. The methods section does not provide information on coefficients and methods mentioned in the Results section, such as the Kaiser-Meyer-Olkin (KMO), scree plot, Cronbach’s alpha, intraclass correlation coefficient, among others. It would be beneficial to elaborate on these in the methods section, along with the acceptable threshold values or interpretations for each.
2. We note that the rates of additional diagnoses in the autism group (57%) is much higher than in the comparison non-autistic group (16%). The authors should give more information about these (n, % of each diagnosis, e.g. in Supplementary material) and, importantly, consider how this might affect their findings. Given the importance of age at diagnosis, it would be good to include more information on this (beyond m and sd in Table A1), including range and the n (%) diagnosed before versus post age 18.
3. Could the authors please explain the use of t-tests for evaluating item discrimination? It may be beneficial to consider item response theory (IRT) methods instead, designed for item-level analysis, which provides a more suitable approach to assess an item's ability to differentiate between individuals with varying levels of autism. We recognise that the initial item reduction was necessary due to the large pool of items, and the sample size might not have been sufficient for an EFA on all 102 items. In addition to considering the mentioned IRT methods, an alternative approach to reduce the number of items could involve assessing test-retest reliability at the item level before proceeding with EFA. This could potentially identify unstable items that may be suitable for removal.
4. Could the authors please provide a reference for considering item loadings of 0.35 as satisfactory? Commonly, literature suggests setting a threshold of 0.4 or 0.5 (Velicer & Fava, 1998; Tabachnick and Fidell, 2013).
5. After the initial item removal before proceeding with factor analysis, there is still a concern if the sample size is sufficient for the analysis given the large pool of items. A widely accepted rule of thumb for factor analysis suggests that 10 cases are needed per indicator variable (Nunnally & Bernstein, 1967 quoted by Wang & Wang, 2012). It would be beneficial for the authors to address this.
6. Could the authors please provide details on the data distribution and address whether there was a presence of missing values, including details on how missing data were handled?
7. Could the authors please provide a rationale for selecting the Primary-Axis Factoring estimator in EFA? Typically, it is employed for continuous data (Kyriazos & Poga-Kyriazou, 2023). However, considering that Weighted Least Squares is specifically designed for ordinal data with fewer than five response options, it might be more appropriate for the current study.
8. In addition to scree plot, we suggest performing parallel analysis for categorical data. This is recommended because scree plots have been demonstrated to potentially overestimate the number of factors in the data (Zwick & Velicer, 1986).
9. Lines 352-353: “The Kaiser-Meyer-Olkin (KMO) measure of sampling accuracy was 0.932, indicating an adequate sample size.” It is important to note that KMO does not evaluate adequacy of sample size but rather suitability of data for factor analysis.
10. Were the intercorrelations and reliability analyses conducted on women only, or were they performed on the entire sample?
11. We recommend conducting multiple group CFA with respect to gender for both AWE and AQ-50. The group-related invariance of the construct is a prerequisite to comparing group means (Putnick & Bornstein, 2016). This would be particularly valuable considering the context presented in the introduction, which raises the possibility of gender bias in the assessment of autism.
12. We were wondering why individuals identifying as other gender were excluded in the sample of autistic participants but not in the general population sample?
Discussion
1. Lines 482-484: “Whilst there was a gender difference on the AQ total score among non-autistic people (men > women), there was no gender difference among autistic people, confirming all earlier studies.”
Could the authors please clarify which earlier studies they are referring to – is it true to say all previous studies show the same pattern?
2. “On the AWE total score, no gender differences were found, either among autistic people or in the general population” (line 484) – should perhaps be qualified by reminding the reader that, in the general population, all subscales were significantly different between men and women.
3. There seems to be a bit of tension between the finding that there are no differences on the AWE between autistic men and women (or general population men and women as mainly emphasized in the Results, e.g., l.394/5), and the aims of the study to capture something specific to the female experience/manifestation of autism. It would be good to see the authors tackle this head on, in the Discussion. At present they sort of dodge the issue by implying their new measure has met their aims because “the AWE covers autistic characteristics that resonate with autistic women” (l. 489).
4. Lines 503-505: “Test-retest reliability was good for both AQ (r = 0.818) and AWE (r = 0.764), showing that the traits measured by both instruments are stable over time.”
It is important to note that test-retest reliability concerns the stability of the measurement tool, not the stability of the trait being measured.
5. Lines 508-511: “Validity of the AWE was higher in women: AWE scores of autistic women were significantly higher than women from the general population, and this difference was greater when examining AWE scores of autistic men compared to men from the general population (2x2 interaction of group x gender).”
It would be more accurate to state that there is evidence supporting the (discriminative) validity of the measure, rather than the validity is higher for one group.
6. Lines 584-585: “This is indicative of gender bias on the item level, which is cancelled out at the test level since no gender effect was found for the total AWE score. Item response theory should be used to further investigate gender bias in the AWE.”
It is important to clarify that differences in scores between the two genders do not necessarily imply a measurement bias of the tool; these are distinct considerations. This could potentially suggest genuine differences in the underlying construct measured by AWE. However, the current study did not conduct gender-related measurement invariance to assess the measurement (non)bias of AWE and if the gender groups can be meaningfully compared. We strongly recommend conducting multiple group CFA to thoroughly investigate this aspect. Furthermore, the source of the claim that gender bias has been cancelled out at the test level is not entirely clear.
7. Lines 619 onwards discuss the correlation between age at diagnosis and the Sensing Boundaries subscale – but in the Results we are told that correcting for age wiped out this correlation. The authors should either unpack why this correction is problematic because there is a logical link between current age and age at diagnosis, or temper their discussion; after all, interoception is known to be affected by the ordinary ageing process (e.g., Nusser, Pollatos & Zimprich, 2021. Age-related effects on interoceptive accuracy, general interoceptive sensibility, and specific interoceptive sensibility. European Journal of Health Psychology. Murphy, et al, 2018. Direct and indirect effects of age on interoceptive accuracy and awareness across the adult lifespan. Psychonomic Bulletin & Review).
8. Lines 667-672: “While a short version of the AQ (AQ-10; Allison et al., 2012; Grove et al., 2017; 667 Hoekstra et al., 2011) is often used to aid referral in primary care settings (for example by general practitioners), these are often limited in internal consistency (and therefore reliability) (Rivet & Matson, 2011). Just one scoring point less or more could make the difference between no referral or referral to specialised care, which may considerably change the course of an individual’s life.”
We would be grateful for clarification whether the authors suggest here that limited internal consistency of a tool can lead to the under or overestimation of scores, subsequently affecting referrals to diagnostic services? If so, this claim should be supported with references. If they mean something else – and the referent of ‘these’ in ‘these are often limited’ is unclear – some rewording might be helpful.
9. In the limitations section, it could be valuable to address concerns related to the sample size for the analysis (please refer to comment 4 under 'Methods and Results') and that the factor structure needs to be further tested via confirmatory factor analysis to confirm the solution derived from EFA.
References
Belcher, H. L., Uglik-Marucha, N., Vitoratou, S., Ford, R. M., & Morein-Zamir, S. (2023). Gender bias in autism screening: Measurement invariance of different model frameworks of the Autism Spectrum Quotient. BJPsych Open, 9(5). https://doi.org/10.1192/bjo.2023.562
Brown, C. M., Attwood, T., Garnett, M., & Stokes, M. A. (2020). Am I Autistic? Utility of the Girls Questionnaire for Autism Spectrum Condition as an Autism Assessment in Adult Women. Autism in adulthood : challenges and management, 2(3), 216–226. https://doi.org/10.1089/aut.2019.0054
English, M. C., Gignac, G. E., Visser, T. A., Whitehouse, A. J., & Maybery, M. T. (2019). A comprehensive psychometric analysis of autism‐spectrum quotient factor models using two large samples: Model recommendations and the influence of divergent traits on total‐scale scores. Autism Research, 13(1), 45–60. https://doi.org/10.1002/aur.2198
Freeth, M., Sheppard, E., Ramachandran, R., & Milne, E. (2013). Across-cultural comparison of autistic traits in the UK, India and Malaysia. Journal of Autism and Developmental Disorders, 43(11), 2569–2583. https://doi.org/10.1007/s10803-013-1808-9
Grove, R., Hoekstra, R. A., Wierda, M., & Begeer, S. (2017). Exploring sex differences in autistic traits: A factor analytic study of adults with autism. Autism : the international journal of research and practice, 21(6), 760–768. https://doi.org/10.1177/1362361316667283
Hoekstra, R. A., Bartels, M., Cath, D. C., & Boomsma, D. I. (2008). Factor structure, reliability and criterion validity of the Autism-Spectrum Quotient (AQ): a study in Dutch population and patient groups. Journal of autism and developmental disorders, 38(8), 1555–1566. https://doi.org/10.1007/s10803-008-0538-x
Hurst, R. M., Mitchell, J. T., Kimbrel, N. A., Kwapil, T. K., & Nelson-Gray, R. O. (2007). Examination of the reliability and factor structure of the Autism Spectrum Quotient (AQ) in anon-clinical sample. Personality and Individual Differences,43(7), 1938–1949. https://doi.org/10.1016/j.paid.2007.06.012
Jia, R., Steelman, Z. R., & Jia, H. H. (2019). Psychometric Assessments of Three Self-Report Autism Scales (AQ, RBQ-2A, and SQ) for General Adult Populations. Journal of autism and developmental disorders, 49(5), 1949–1965. https://doi.org/10.1007/s10803-019-03880-x
Kloosterman, P. H., Keefer, K. V., Kelley, E. A., Summerfeldt, L. J., & Parker, J. D. A. (2011). Evaluation of the factor structure of the autism-spectrum quotient. Personality and Individual Differences, 50(2), 310–314. https://doi.org/10.1016/j.paid.2010.10.015
Kyriazos, T. and Poga-Kyriazou, M. (2023) Applied Psychometrics: Estimator Considerations in Commonly Encountered Conditions in CFA, SEM, and EFA Practice. Psychology, 14, 799-828. doi: 10.4236/psych.2023.145043.
Lau, W. Y.-P., Gau, S. S.-F., Chiu, Y.-N., Wu, Y.-Y., Chou, W.-J., Liu, S.-K., & Chou, M.-C. (2013). Psychometric Properties of the Chinese version of the Autism Spectrum Quotient (AQ). Research in Developmental Disabilities, 34(1), 294–305. https://doi.org/10.1016/j.ridd.2012.08.005
Murray, A. L., Booth, T., Auyeung, B., McKenzie, K., & Kuenssberg, R. (2019). Investigating Sex Bias in the AQ-10: A Replication Study. Assessment, 26(8), 1474–1479. https://doi.org/10.1177/1073191117733548
Nunnally, J. C., & Bernstein, I. H. (1967). Psychometric Theory (Vol. 226). New York, NY: McGraw-Hill.
Putnick, D. L., & Bornstein, M. H. (2016). Measurement Invariance Conventions and Reporting: The State of the Art and Future Directions for Psychological Research. Developmental review: DR, 41, 71–90. https://doi.org/10.1016/j.dr.2016.06.004
Tabachnick, B. G., & Fidell, L. S. (2013). Using Multivariate Statistics (6th ed.). Boston, MA: Pearson.
Taylor, E. C., Livingston, L. A., Clutterbuck, R. A., & Shah, P. (2020). Psychometric concerns with the 10-item autism-spectrum quotient (AQ10) as a measure of trait autism in the general population. Experimental Results, 1. https://doi.org/10.1017/exp.2019.3
Velicer, W. F., & Fava, J. L. (1998). Affects of variable and subject sampling on factor pattern recovery. Psychological Methods, 3(2), 231–251. https://doi.org/10.1037/1082-989X.3.2.231
Wang, J., & Wang, X. (2012). Structural Equation Modeling: Applications Using Mplus. Hoboken, NJ: Wiley, Higher Education Press. https://doi.org/10.1002/9781118356258
Zwick, W. R., & Velicer, W. F. (1986). Comparison of five rules for determining the number of components to retain. Psychological Bulletin, 99(3), 432–442. https://doi.org/10.1037/0033-2909.99.3.432
Comments on the Quality of English LanguageJust a couple of small typos, otherwise very good.
